# Gender Differences and Comorbidities in U.S. Adults with Bipolar Disorder

**DOI:** 10.3390/brainsci8090168

**Published:** 2018-09-01

**Authors:** Rikinkumar S. Patel, Sanya Virani, Hina Saeed, Sai Nimmagadda, Jupi Talukdar, Nagy A. Youssef

**Affiliations:** 1Department of Psychiatry, Griffin Memorial Hospital, 900 E Main St, Norman, OK 73071, USA; 2Department of Psychiatry, Maimonides Medical Center, 4802 10th Ave, Brooklyn, NY 11219, USA; svirani@maimonidesmed.org; 3Baqai Medical University, 51, Deh Tor, Gadap Road, Super Highway, Karachi 74600, Pakistan; hinasadeel@gmail.com; 4University of Illinois College of Medicine, 1 Illini Dr, Peoria, IL 61605, USA; nimmagaddasaik@gmail.com; 5Gauhati Medical College and Hospital, GMCH Rd, Bhangagarh, Guwahati, Assam 781032, India; jupi1589@gmail.com; 6Medical College of Georgia, Augusta University, 997 St. Sebastian Way, Augusta, GA 30912, USA; NYOUSSEF@augusta.edu

**Keywords:** bipolar disorder, comorbidities, gender differences, inpatient psychiatry

## Abstract

Background: Past studies have evaluated the association of various comorbidities with bipolar disorder. This study analyzes differences in the prevalence and association of medical and psychiatric comorbidities in bipolar patients by gender. Methods: A retrospective analysis was conducted using the Nationwide Inpatient Sample (2010–2014). Using International Classification of Diseases, 9th Revision, Clinical Modification (ICD-9-CM) codes, we narrowed the study population to comprise those with a primary diagnosis of bipolar disorder and then obtained information about comorbidities. The differences in comorbidities by gender were quantified using chi-square tests and the logistic regression model (odds ratio (OR)). Results: Hypertension (20.5%), asthma (12.5%) and hypothyroidism (8.1%) were the top medical comorbidities found in bipolar patients. Migraine and hypothyroidism were seen three times higher in females (OR = 3.074 and OR = 3.001; respectively). Females with bipolar disorder had higher odds of comorbid inflammatory disorders like asthma (OR = 1.755), Crohn’s disease (OR = 1.197) and multiple sclerosis (OR = 2.440) compared to males. Females had a two-fold higher likelihood of comorbid post-traumatic stress disorder (PTSD) (OR = 2.253) followed by personality disorders (OR = 1.692) and anxiety disorders (OR = 1.663) compared to males. Conclusion: Women with bipolar disorder have a much higher medical comorbidity burden than men and may highly benefit from an integrated team of physicians to manage their condition and improve their health-related quality of life.

## 1. Introduction

Bipolar disorder is a prevalent and chronic psychiatric illness that can be disabling if not well treated. It is commonly associated with comorbid systemic illnesses, morbidity, functional impairment and a high risk of suicide and mortality [1]. With proper diagnosis and treatment, patients can live productive and largely healthy lives. Although bipolar and unipolar disorders differ in some aspects, including treatment, they share common symptomatic and functional impairments, especially in depressive episodes. These disorders have some shared impairments in the white and grey matter compartments on brain imaging studies, but more white matter abnormalities have been reported in bipolar disease than in unipolar disorders. Brain imaging has been used to find the structural integrity of cortical white matter and grey matter [2]. Distinctive abnormalities of white matter connectivity and emotional and sensory neural circuitry are noted in bipolar disorders. These white matter abnormalities may be related to both axonal disorganization and demyelination or apoptosis [2].

Medical and psychiatric comorbidities of bipolar disorder have been a topic of increasing concern for healthcare providers, as patients with bipolar disorder may be at an increased risk of certain medical and other psychiatric conditions, which is the focus of this paper [3].

Though the rationale for the frequent existence of medical comorbidities in patients with bipolar disorder compared to other psychiatric illnesses is not fully understood, associations have been explored and explained by common risk factors, bidirectional causal factors and/or a common biological basis shared with certain medical conditions [4]. Increased comorbidities can be attributed to the genetic basis of the disorder. There seems to be some common genetic risk with conditions such as Alzheimer’s disease, diabetes and coronary heart disease [5]. A general model of the genetics has been proposed that emphasizes the shared nature of common alleles in related common conditions, such as schizophrenia and bipolar disorder, diabetes and autoimmune diseases [5]. It has been shown that there is a substantial medical burden associated with bipolar disorder [6]. These patients can greatly benefit from an integrated team of healthcare providers [6]. 

The objective of this study is to analyze and discern the differences and burden of medical and psychiatric comorbidities in bipolar patients by gender. This is the first study using the Nationwide Inpatient Sample (NIS) data that explores the prevalence of various comorbidities in bipolar disorder in adult inpatients. 

## 2. Methods

### 2.1. Data Source

A retrospective, cross-national, population-based analysis was performed using the Healthcare Cost and Utilization Project’s (HCUP) NIS data from the years 2010 to 2014 [7]. The Agency for Healthcare Research and Quality (AHRQ) sponsors the HCUP databases, which are specifically designed to determine hospital outcomes and comorbidities related to the disease of interest. The HCUP-NIS database is the largest inpatient database comprising the administrative records from 4411 hospitals and covering 45 states in the United States [7]. The large sample size available via the database facilitated further recognition and analyses of rare comorbidities and special patient populations.

### 2.2. Selection of Patients

Based on the International Classification of Disease, Ninth Revision Clinical Modification (ICD-9-CM) diagnosis codes, we identified patients 18–50 years old with a primary diagnosis of bipolar disorder at the time of inpatient admission. In HCUP databases, more than 14,000 ICD-9-CM diagnosis codes have been mentioned. Bipolar disorder was identified using diagnosis codes 296.40, 296.41, 296.42, 296.43, 296.44, 296.50, 296.51, 296.52, 296.53, 296.54, 296.60, 296.61, 296.62, 296.63, 296.64 and 296.7. 

### 2.3. Variables of Interest

The demographic variables examined in this study included age group (18–50 years), gender (male or female) and race (Caucasian, African American, Hispanic and other). Comorbidities were considered coexisting conditions to bipolar disorder, which is the primary disorder under this study [7]. Using ICD-9-CM codes, this variable identified various medical and psychiatric comorbidities in the patient records, as mentioned in Table 1. 

### 2.4. Approaches

This study was performed using the HCUP-NIS database, focusing on the determination of the comorbidities for patients with bipolar disorder. Descriptive statistics and cross tabulation were used to summarize the results. Pearson’s Chi-square test and independent sample *t*-test were used for categorical data and continuous data, respectively. We used a logistic regression model to measure the associations between females and males (reference category) in terms of medical and psychiatric comorbidities. We applied discharge weights in all regression models to obtain nationally representative inpatient data, whereas age and race were adjusted. A *p* value < 0.01 was used as a reference to determine the statistical significance test result. All statistical analyses were done by SPSS version 23 (International Business Machines Corporation, Armonk, NY, USA) in this study [8].

### 2.5. Ethical Approval

Our study database did not contain patient identifiable information. To protect the privacy of individual patients, physicians and hospitals, the state and hospital identifiers were all de-identified; this database is available as a public database. The use of this administrative database under the HCUP, according to the Agency for Healthcare Research and Quality (AHRQ) of the U.S. Department of Health and Human Services, does not require approval from an Institutional Review Board (IRB), because the NIS is publicly available and is a fully de-identified database. 

## 3. Results

### 3.1. Sample Characteristics

We analyzed 593,257 patients admitted for bipolar disorder from 2010 to 2014; 53.3% were young adults (18–35 years of age) and 54.8% were females. About two-thirds of the sample population was Caucasian (69.8%) followed by African American (16.8%) and Hispanic (8.4%). 

### 3.2. Medical and Psychiatric Comorbidities

Hypertension (20.5%), asthma (12.5%) and hypothyroidism (8.1%) were the top three medical comorbidities found in bipolar patients. Migraine and hypothyroidism were seen three times higher in females (OR = 3.074 and OR = 3.001; respectively). Females with bipolar disorder had higher odds of comorbid inflammatory disorders like asthma (OR = 1.755), Crohn’s disease (OR = 1.197) and multiple sclerosis (OR = 2.440) compared to men. Among the cardiometabolic comorbidities, females had higher odds of obesity (OR = 2.011) followed by diabetes (OR = 1.159). The gender-wise distribution of comorbidities in patients with bipolar disorder is shown in Figure 1. 

Among substance use disorders, 50.8% of males with bipolar disorder had a co-diagnosis of drug abuse and 28.5% had alcohol abuse. On the contrary, females had lower odds of these substance-use disorders than males. The most common psychiatric illness associated with bipolar disorder was anxiety disorder (24.1%) followed by personality disorder (17.5%) and PTSD (9.7%). Females had twice the odds of comorbid PTSD (OR = 2.253) followed by personality disorder and anxiety disorder (OR = 1.692 and OR = 1.663; respectively). Although eating disorders were seen in very low proportions of bipolar patients (0.1% males and 0.6% females), females had about 11 times higher odds of this comorbidity compared to men (OR = 11.673). The association of comorbidities in females is shown in Table 2. 

## 4. Discussion

This study of population-based hospital data from patients with bipolar disorder reveals the association with various medical and psychiatric comorbidities and gender differences. Hypertension was the most common comorbidity in bipolar patients (20.5%), but was lower when compared to the general global population, as mentioned in a systematic study from 2000–2010 (28.5% in high-income countries and 31.5% in low- and middle-income countries) [9] and the recent report stating that hypertension is seen in 30.2% of men and 27.7% of women in the general population [10]. Many psychotropic medications such as tricyclic antidepressants can induce hypotension, which can mask the symptoms of hypertension during routine health check-ups. Thus, the number of hypertensive patients with bipolar disorder could potentially be lower [11]. Another possible reason for the lower number of bipolar patients with hypertension could be due to fewer follow-up visits of mental and behavioral patients to the primary health clinics [12]. As per the Global Asthma Report, about 8.6% of young adults (age 18–45) experienced the symptoms of asthma in 2014, whereas comorbid asthma was seen in a higher proportion of bipolar patients (12.5%) in our study [13]. Hypothyroidism was seen in 8.1% bipolar patients in our study population, which was two times higher than that seen in the general US population (4.6%) as per the recent report by the Endocrine Society [14]. Bipolar disorder has been found to have strong affiliations with thyroid dysfunction, whether the interaction of lithium is considered or not [15,16]. Previous studies have provided evidence linking hypothyroidism with bipolar disorder, with a notable association with manic relapse [17] and rapid cycling bipolar disorder [18]. Our data results also showed an increased association of thyroid dysfunction with bipolar disorder. Hypothyroidism was the third most common medical comorbidity in bipolar patients. Also, females with bipolar disorder in this study had a three-fold higher likelihood of comorbid hypothyroidism compared to males. Migraines also had a remarkable association in females with bipolar disease in our study; they were three times more likely to have migraines as a comorbidity than males. There exists a higher percentage of the co-occurrence of migraines and bipolar disorders, predominantly in subjects with a confirmed family history of bipolar disorder, suicidal attempts and childhood physical abuse [19]. In addition to the higher prevalence ranging from 30 to 34.8% [20,21], patients with bipolar disorders and migraines were also found to have a complex course of the disease with more severe and an increased number of depressive episodes, as well as suicidality [22]. Oxidative stress and inflammation in cross sensitization between bipolar disorder and migraine has been hypothesized as a possible explanation [22]. Genetic factors are also thought to be implicated in the co-occurrence of migraine and bipolar disorders. In addition, inflammatory mechanisms involving abnormal cytokine activation and abnormal arachidonic acid metabolism have also been postulated as a common pathogenesis [23]. Both of these disorders also demonstrated higher neuroticism, which might, in part, explain the co-occurrence [23]. Similar to that seen in our study, the prevalence of migraine is also higher in females in the general population than males (19.1% vs. 9%) [24], though this comorbidity has a lower overall prevalence compared to that present in bipolar patients (8.4% in females and 2.8% in males).

Bipolar disorder has been associated with many autoimmune disorders and few studies have indicated the coexistence of multiple sclerosis [25,26,27]. The involvement of the *human leukocyte antigen (HLA)* gene complex has been postulated as one of the possible explanations for this increased susceptibility [24]. Analysis of our data also suggests a possible linkage of bipolar and multiple sclerosis. Females with bipolar disorder were found to have about three times the likelihood of having multiple sclerosis (as a comorbidity) compared to males. Some studies have shown that bipolar disorder is highly prevalent in patients with asthma and, therefore, suggest a further increased risk with the use of a higher prednisone dose [28,29]. Potential shared genetic vulnerability has also been proposed [30] as a possible explanation for this association. However, no precise etiology for this co-occurrence has been established. The female participants in our study were about two times more likely to have comorbid asthma compared to the male participants. Depression and anxiety disorders have been more frequently reported to be associated with inflammatory bowel disease [31], but previous studies were not able to establish a strong association of bipolar disorder with Crohn’s disease. Nevertheless, we found a marginally higher likelihood of a co-diagnosis of Crohn’s disease in females than males.

Metabolic derangements resulting in obesity and diabetes are commonly reported with bipolar disorder [32,33]. Many mechanisms have been suggested for these findings. Impaired glucose metabolism, poor quality of life and commonly associated eating disorders are implicated in these dysregulations [34,35]. Women with bipolar disorder, when compared to men with bipolar disorder, have higher rates of abdominal obesity as per a systematic review conducted by Baskaran et al. [36]. As per the National Diabetes Statistics Report (2017), diabetes is seen in 14.9% of women and 15.3% of men, which is much higher than that seen in the bipolar patients in this study [37]. Nonetheless, bipolar females had a higher risk of obesity and diabetes than the male participants in our study. The National Health Interview Survey 1997–2012 concluded that comorbid obesity is present in 38.3% of women and 34.3% of men [38], which is about three times higher than that seen in the bipolar patients in our study. Unlike the above reports, diabetes and obesity seem to be lower in the bipolar inpatient population, yet the odds of these comorbidities were higher in females with bipolar disorder compared to males.

Among all of the listed psychiatric comorbidities, female patients with bipolar disorder were found to be most vulnerable to having post-traumatic stress disorder (PTSD). These results are consistent with the findings of previous studies that indicated the co-occurrence of PTSD with bipolar disorder at significantly higher odds [39,40]. These studies revealed a significant co-occurrence of PTSD with bipolar disorder in both veterans and the civilian population. The bipolar patients had a higher prevalence of comorbid PTSD in our study compared to that seen in the general population. A recent National Institute of Mental Health (NIMH) survey states that PTSD is present in 18% of males and 5.2% of females [41], and personality disorder is present in 9.1% of the general population [42]. The females with bipolar disorder in our study showed a 1.7 times higher association with personality disorders and anxiety disorder than males with bipolar disorder. Among the personality disorders, some similarities have been found between bipolar disorder and borderline personality disorder, although there are still many differences. Dysfunction of the dopaminergic and serotonergic systems, changes in limbic system, as well as the size of the amygdala have been proposed in the pathophysiology of both disorders [43]. 

Past studies have found a strong association of anxiety disorder with bipolar disorder [44,45]. We found that females with bipolar disorder were 1.6 times more likely to suffer from anxiety disorder compared to males. When compared with the general population, bipolar patients had a higher prevalence of comorbid anxiety in males (14.3% vs. 19.1% seen in bipolar patients) and females (23.4% vs. 28.2% seen in bipolar patients) [46]. An eating disorder was present in a very low proportion of total patients (0.1% males and 0.6% females), which is lower than that seen in the general population (0.4% males and 1% females) [47]. However, the females with bipolar disorder had 11-fold higher odds of having eating disorders with bipolar disorder than males in our study. There is an increased coexistence of bipolar disease with eating disorders in female patients, predominantly bulimia and anorexia nervosa [48,49]. In addition, the coexistence of bipolar disorder with an eating disorder has been reported to be associated with poor long-term outcomes and a severe disease course of bipolar disorder [49,50]. Alcohol use disorder and substance use disorder were seen in 8.4% and 12.5% of men, and 4.2% and 7.9% of women, respectively, in the general population [51,52], which is much lower than what is seen in the inpatient bipolar population in our study.

Furthermore, our study did have some limitations. Firstly, this study was based on an administrative database and lacks patient-level clinical information. NIS data regarding bipolar disorder are limited to inpatient hospitalization only and do not include any data from outpatient settings. Re-hospitalizations, which add to the total inpatient burden, are not accounted for in our study, given the nature of the data. Selection bias is possible, given the retrospective nature of the study. The prevalence of comorbidities in the study participants may differ than that seen in the general population and other bipolar populations, as our participants were selected from the hospital admissions and the inpatient database. However, despite these limitations, NIS is still an excellent inpatient population-based representation of disease associations and comorbidities. Despite the retrospective nature of the study, the chances of recall bias are probably minimal, given that it is an administrative database with primary and secondary diagnosis codes and other clinical information collected at the time of inpatient management, as well as on discharge. The main strength of this study is the nationally representative sample provided by the NIS dataset [7], as well as a uniform collection of data obtained over five years through ICD-9-CM diagnosis codes. Another important strength is its large sample size of 593,257 and the reliability of the data, given that the information is coded independently of the individual practitioner; therefore, this would minimize reporting bias, as the large sample size increases power in detecting differences.

## 5. Conclusions

Through this retrospective nationwide study, we found that women with bipolar disorder have a greater risk and comorbidity with autoimmune and inflammatory disorders than men. Medical and psychiatric comorbidities can substantially add to impairment in function if not treated. These psychiatric comorbidities include anxiety disorders, PTSD, eating disorders and personality disorders. PTSD was the most common psychiatric co-morbidity in bipolar patients. These added comorbidities can substantially increase the disease burden of patients with bipolar disorder. Women with bipolar disorder are at an even more increased risk than men and integrating care among primary care physicians and psychiatrists closely can greatly improve the health-related quality of life and vastly decrease the burden of illness and complications. This can be attained by closer and more frequent checks and follow-ups for medical comorbidities for female patients with bipolar disorder, along with aggressive control of the physical comorbidities. During the follow-up visits, the patients should be educated by providing clear, written instructions regarding health education and directions. Frequent bipolar psychiatric comorbidities like PTSD and substance use or alcohol use disorders are either undetected or under-treated. We recommend thorough psychiatric interviewing and necessary laboratory tests for early diagnosis and treatment of these comorbidities. This could improve the overall psychiatric care for patients with bipolar disorder. Appropriate and regular communication between the primary care physicians and psychiatrists managing the bipolar patients is required in order to maximize integration and provide high-quality health care. We recommend an integrated care model, involving psychiatrists and primary care physicians to improve treatment of both the psychiatric and medical comorbidities in these patients.

## Figures and Tables

**Figure 1 brainsci-08-00168-f001:**
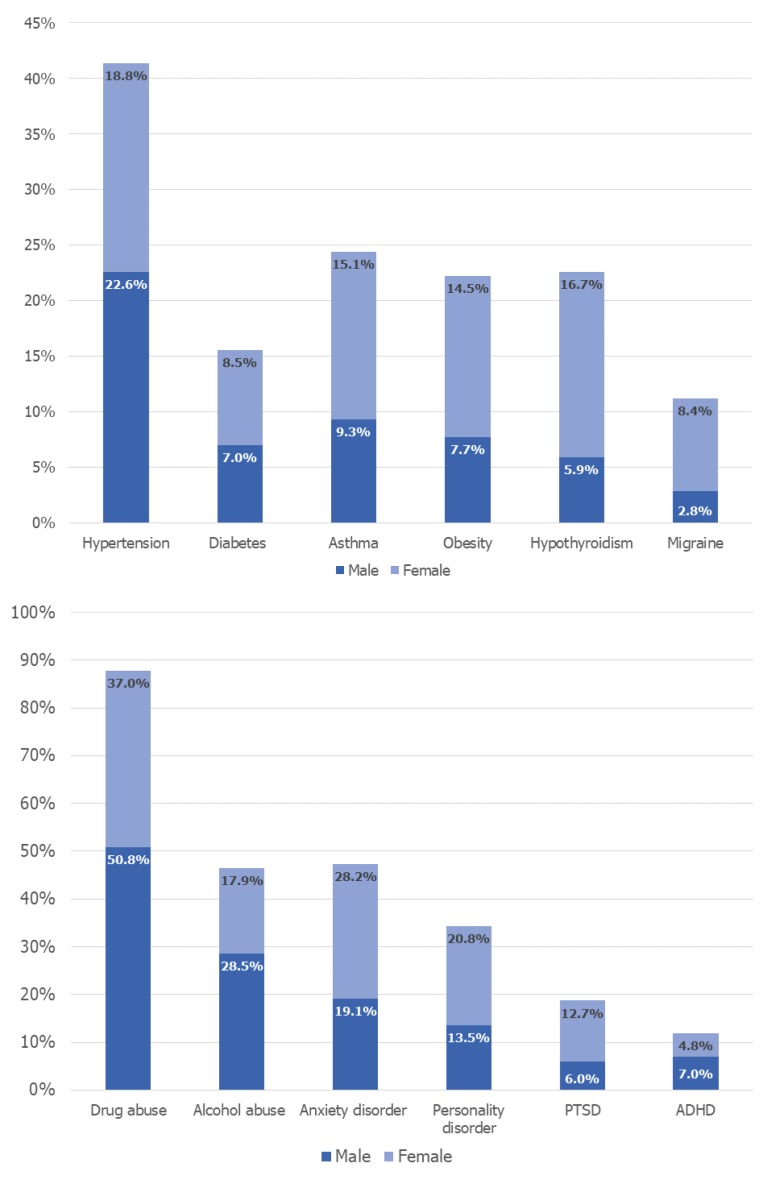
Gender-wise distribution of comorbidities in patients with bipolar disorder. The proportion of males and females with bipolar disorder were obtained using cross tabulation and the Pearson Chi-Square (χ^2^) test, and were significant with *p* value ≤ 0.01 at a 95% confidence interval. PTSD: Post-traumatic stress disorder; ADHD: Attention-deficit/hyperactivity disorder.

**Table 1 brainsci-08-00168-t001:** International Classification of Disease, Ninth Clinical Modification (ICD-9-CM) codes used to identify comorbidities in patients with bipolar disorder.

Comorbidity	ICD-9-CM Diagnosis Code
Medical Comorbidities	
Hypertension	401.1, 401.9, 642.00–642.04, 401.0, 402.00–405.99, 437.2, 642.10–642.24, 642.70–642.94
Diabetes	249.00–249.31, 250.00–250.33, 648.00–648.04
Obesity	278.0, 278.00, 278.01, 278.03, 649.10–649.14, 793.91, V85.30–V85.39, V85.41–V85.45, V85.54
Hypothyroidism	243–244.2, 244.8, 244.9
Migraine	339.00–339.05, 339.09–339.12, 339.20–339.22, 339.3, 339.41–339.44, 339.81–339.85, 339.89, 346.0, 346.00–346.03, 346.1, 346.10–346.13, 346.2, 346.20–346.23, 346.30–346.33, 346.40–346.43, 346.50–346.53, 346.70–346.73, 346.8, 346.80–346.83, 346.9, 346.90–346.93, 784.0
Crohn’s Disease	560.89, 560.9
Multiple Sclerosis	340
Asthma	493.00–493.02, 493.10–493.12, 493.20–493.22, 493.81, 493.82, 493.90–493.92
Psychiatric Comorbidities	
Drug abuse	292.0, 292.82–292.89, 292.9, 304.00–304.93, 305.20–305.93, 648.30–648.34
Alcohol abuse	291.0–291.3, 291.5, 291.8, 291.81, 291.82, 291.89, 291.9, 303.00–303.93, 305.00–305.03
Anxiety disorder	293.84, 300.00–300.02, 300.09, 300.10, 300.20–300.23, 300.29, 300.3, 300.5, 300.89, 300.9, 308.0–308.4, 308.9, 309.81, 313.0, 313.1, 313.21, 313.22, 313.3, 313.82, 313.83
Personality disorder	301.0, 301.10–301.13, 301.20–301.22, 301.3, 301.4, 301.50, 301.51, 301.59, 301.6, 301.7, 301.81–301.84, 301.89, 301.9
PTSD	309.81
ADHD	314.00, 314.01
Eating disorder	307.1, 307.50–307.54, 307.59

PTSD, Post-traumatic stress disorder; ADHD: Attention-deficit/hyperactivity disorder.

**Table 2 brainsci-08-00168-t002:** Odds of medical and psychiatric comorbidities with bipolar disorder in females compared with males.

Comorbidities	Logistic Regression Model		
	OR	95% CI	*p*
Medical Comorbidities			
Hypertension	0.721	0.711–0.731	<0.0001
Diabetes	1.159	1.136–1.182	<0.0001
Obesity	2.011	1.976–2.046	<0.0001
Hypothyroidism	3.001	2.936–3.069	<0.0001
Migraine	3.074	2.995–3.155	<0.0001
Asthma	1.755	1.726–1.783	<0.0001
Crohn’s disease	1.197	1.108–1.293	<0.0001
Multiple Sclerosis	2.440	2.183–2.727	<0.0001
Psychiatric Comorbidities			
Drug abuse	0.573	0.567–0.579	<0.0001
Alcohol abuse	0.529	0.523–0.536	<0.0001
Anxiety disorder	1.663	1.643–1.684	<0.0001
Personality disorder	1.692	1.668–1.716	<0.0001
PTSD	2.253	2.211–2.296	<0.0001
ADHD	0.699	0.683–0.714	<0.0001
Eating disorder	11.673	9.914–13.744	<0.0001

Significant *p* values ≤ 0.01 at 95% Confidence Interval. Odds Ratio generated by binomial logistic regression model and was adjusted for age and race. OR: Odds ratio; CI: Confidence interval; PTSD: Post-traumatic stress disorder; ADHD: Attention-deficit/hyperactivity disorder.

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
