# Peer review of "Gender Differences and Comorbidities in U.S. Adults with Bipolar Disorder"

_brainsci, 2018, doi:10.3390/brainsci8090168_

Round 1

Reviewer 1 Report

This is, in summary, a retrospective study aimed to analyze the differences in prevalence and association of medical and psychiatric comorbidities in bipolar patients by gender. The authors found that hypertension (20.5%), asthma (12.5%), and hypothyroidism (8.1%) were the the most frequent medical comorbidities found in bipolar patients. Migraine and hypothyroidism were observed three times higher in females. Females with bipolar disorder had higher odds of comorbid inflammatory disorders like asthma, Crohn’s disease and Multiple Sclerosis compared to males. In addition, females had two-fold higher likelihood of comorbid PTSD  followed by personality disorder and anxiety disorder than males.

The authors found as follows my main comments/suggestions.

First, when, within the Introduction section, the authors correctly referred to the chronic and disabling nature of bipolar disorder which is frequently associated with impaired psychosocial functioning, they could also mention that both bipolar disorder (BD) and unipolar disorder (UD) share common symptomatic and functional impairments. Various brain imaging techniques have been used to investigate the integrity of brain white and gray matter in these disorders. UD and BD present both shared and distinctive impairments in the white and grey matter compartments but more white matter abnormalities have been reported in bipolar disease than in unipolar disease.. In order to address this issue, i suggest to cite and briefly discuss within the main text, the paper published in 2014 on Eur Child Adolesc Psychiatry (PMID: 25212880).

In addition, as the authors well described the main aims/objectives of this study, the most relevant hypotheses of this paper may be extensively reported as well.

Furthermore, when, throughout the Methods section, the authors reported that this is a cross-national population-based analysis, here they should specify that this is really a retrospective study which may be associated to a recall bias. The authors are required to focus, at least briefly, on this issue.

Moreover, as the authors referred, within the Discussion section, to the higher percentage of co-occurrence of migraine and bipolar disorder, predominantly in subjects with a positive family history of bipolar disorder, suicidal attempts, and childhood physical abuse, they should provide the major hypothesized determinants underlying this link. Importantly, how bipolar individuals with positive family history of bipolar, suicidal attempt and childhood physical abuse may be more likely to develop migraine and presumably an impaired quality of life?

Also, what is the take-home message of this paper? While the authors reported that women with bipolar disorder are at even more increased risk than men and may highly require an integrated team of physicians to manage their condition and improve the health-related quality of life, they failed, in my opinion, to provide some conclusive remarks upon the main topic for the general readership. Specifically, how the integrated team of physicians may better manage women with bipolar disorder? What are, according to the authors’ expertise, the most relevant management strategies to this regard? Here, more details/information are needed.

Finally, the manuscript needs to be reviewed by a native English speaker for the quality of language.

Author Response

Hello,

Thanks for reviewing our manuscript. Your comments were valuable in order to make the paper more interesting for the readers. Have addressed all the points and will submit the revised file.

The retrospective type has been added to methods, but the chances of recall bias is minimal as its an administrative data and the same is mentioned in limitations. 

Reviewer 2 Report

This is a vague and unfocused  manuscript reporting medical and psychiatric comorbidities of inpatient admissions with a primary diagnosis of bipolar disorder, using as source the Nationwide Inpatient Sample.

Suffice to say that Authors announce a Table 2 which is not provided, that the manuscript has typos such as fiction instead of function, that the paper oscillates continuosly between considering the prevalence of various conditions in this population and in other not comparable populations and comparing rather random odds ration between females and males in the study population.

It is unclear how the comorbid conditions reported have been selected, unclear why migraine is classified as an autoimmune disorder....

But, most of all, Authors seem not take in consideration the different gender distribution of the comorbid conditions studied in influencing their outcome 

Author Response

Thanks for reviewing our manuscript. Here are my answers to your comments.

The study is focussed in determining the prevalence of medical and psychiatric comorbidities in inpatient population with admitting diagnosis of bipolar disorder. We have also evaluated the odds ratio of having these comorbidities in females compared to males. Table 2. was included in the manuscript and somehow the journal editors have reformatted the paper and may have forgotten to add table 2. The editors can better address this as they have the original file where table 2 was included. However, we will submit the revised file and include table 2 again.

Typos such as fiction instead of function has been corrected as well as other minor errors. As  the main aim of the study was to evaluate the prevalence and the gender differences so the paper describes the same.

Changes have been made in the discussion related to migraine section along with more clear explanations.

Authors have well taken into consideration the different gender distribution of the comorbid conditions and depicted the same in terms of prevalence in the figure and the risk of comorbidities in females as shown in Table 2, which you were not able to visualize due to error from journal editing issue and not the authors of the paper. 

Appreciate your valuable time and recommendations made to make our paper stronger. Thanks

Round 2

Reviewer 1 Report

In the revised manuscript, the authors addressed most of the major questions raised by Reviewers improving both the main structure and quality of this paper. I have no further additional comments.  

Author Response

Thank you for the time and concern. We have already checked the manuscript for English and grammatical errors before prior resubmission. Hope that helps.  

Reviewer 2 Report

The new version of the manuscript is indeed improved. 

My understanding is that the submitting Authors needs to pre-approve the .pdf for final submission so blaming the journal editors for the lack of Table 2 in the first version of the manuscript sounds a bit odd to me.

You have not addressed at all my concern about the fact that many of the comorbid conditions that you report have an uneven gender distribution by themselves. Therefore main questions about comorbid conditions in bipolar disorder, e.g.  if these comorbid conditions gender ratio reflect those of the general population or are peculiar for bipolar disorder, or even if these comorbidities ara more likely in bipolar disorder vs all other combined primary diagnoses, remain unanswered. 

Comparing overall comorbidities rates with those found in bipolar disorder patients of very different populations makes little sense to me, at least the limitations of this approach should be mentioned (as already remarked)

Also, the method section should be more clear regarding the issue of readmissions, that is if the analysis concerns 593,257 patients or 593,257 admission episodes, giving a few more details on the characteristics of the database

Author Response

Our team apologizes for any inconvenience caused due to the lack of provision of a full-text manuscript (table 2. was missing) due to human and software errors. Comment 1: The comorbid conditions gender ratio reflect those of the inpatient with a primary diagnosis of bipolar disorder only. We did explain these gender differences in the discussion part. We have nowhere talked about its relationship with the general population as our focus was only on bipolar patients so it doesn't make sense to our team. The answer to comment 2 answered as below and edited in limitations: The prevalence of comorbidities in the study participants may be higher than that seen in the general population as our participants were selected from the hospital admissions and the inpatient database of NIS from 2010 to 2014. The prevalence of comorbidities in the study participants may be higher than that seen in the general population as our participants were selected from the hospital admissions and the inpatient database of NIS from 2010 to 2014. Comment 3: Characteristics of the database is well mentioned in the Methods and also the query about readmissions is answered in the limitation section since the first day of submission.